# Status Epilepticus and Neurosyphilis: A Case Report and a Narrative Review

Giada Giovannini [1,2,*] and Stefano Meletti [1,3]

1 Neurology Unit, OCB Hospital, Azienda Ospaliera-Universitaria, 41126 Modena, Italy; stefano.meletti@unimore.it
2 PhD Program in Clinical and Experimental Medicine, University of Modena and Reggio Emilia, 41126 Modena, Italy
3 Department of Biomedical, Metabolic and Neural Science, University of Modena and Reggio Emilia, 41126 Modena, Italy
* Correspondence: giovannini.giada@gmail.com; Tel.: +39-059-3961676; Fax: +39-059-3962409

**Abstract:** Neurosyphilis is a rare but life-threatening complication of syphilis that can develop even decades after the primary infection and can be unrecognized. Seizures and status epilepticus (SE) may represent the first manifestation in a previously undiagnosed syphilitic patient. We present an exemplification case of a new onset refractory status epilepticus caused by neurosyphilis and we reviewed the existing literature. We selected all studies reporting cases of SE in the context both of patients with a known diagnosis of syphilis and as the first manifestation of neurosyphilis. We identified 50 patients, mostly composed of immunocompetent, middle-aged males. Thirty-nine patients (83%) presented a new onset SE. A history of subtle and rapidly progressive mood and/or cognitive impairment suggesting a limbic encephalitis-like presentation was frequently observed. Focal frontal or temporal SE was reported in 26. Brain MRI frequently showed T2/FLAIR hyperintensities widely involving the medial temporal structures and the frontal lobes. This review should increase the clinician's awareness of neurosyphilis as a possible etiology of a new onset SE of unknown etiology, especially in the context of a "limbic encephalitis"-like clinical presentation. Prompt recognition and treatment for neurosyphilis partially or completely reverse neurologic sequelae, changing the natural history of the disease.

**Keywords:** status epilepticus; new onset refractory status epilepticus; syphilis; neurosyphilis; lue; etiology

## 1. Introduction

*Syphilis*, widely known as "the great imitator" for the multitude of its different and subtle clinical presentations, is an infectious venereal disease caused by the spirochete bacterium *Treponema pallidum* subspecies *pallidum.* After the advent of penicillin and the development of public health care measurements, the incidence of syphilis was reduced, but since the end of the 20th century it has started to increase in parallel with HIV infection with which syphilis has a synergistic link [1,2]. Syphilis can be acquired by person-to-person transmission (where organisms enter via skin micro-abrasion or mucous membranes) but also through blood transfusion, solid organ transplant or vertically. The primary syphilis is generally characterized by the appearance of a solitary non-tender chancre generally affecting genital areas (but other mucosae could be affected as well) together with local lymphadenopathy. If untreated, after the first local invasion, T. pallidum rapidly disseminates systemically by binding to endothelial cells and crossing vascular barriers through tight junctions. The hematogenous dissemination brings out the clinical protean manifestations of secondary syphilis (condyloma lata, hand and foot lesions, macular rash, diffuse lymphadenopathy, headache, myalgia, arthralgia, pharyngitis, hepatosplenomegaly, alopecia and malaise). Primary and secondary syphilis can resolve

without treatment or with the patient entering a latent phase in which syphilitic serologic tests are positive but there are no clinical manifestations of the disease [1]. Patients entering the tertiary stage can develop systemic serious manifestations represented by cardiovascular syphilis and gummous syphilis as well as late *neurosyphilis* [2].

CNS spirochetal invasion can happen within days after the primary infection. If an inflammatory response to CNS invasion does not occur, a spontaneous resolution happens. Otherwise, if an inflammatory response to CNS invasion occurs, the patient develops asymptomatic neurosyphilis. The infection can remain subclinical for many years or even decades, however, these patients are at risk of converting to a symptomatic form. The immune response toward T. pallidum is Th1-predominant, based on CD4+ and interferon $\gamma$. If the immune-mediated clearance of T. pallidum from the CNS becomes defective, neurosyphilis can become symptomatic, therefore immune-depressed patients (such as HIV patients) are at high risk [3].

Neurosyphilis can show different clinical manifestations based on the time from the first infection.

Currently, the yearly incidence of symptomatic neurosyphilis has been estimated at about 0.2–2.1 cases per 100,000 inhabitants [4].

From the classification point of view, neurosyphilis can be divided into early and late types.

In the early neurosyphilis, two different forms have been described:

-   asymptomatic form: CSF abnormalities are found in a patient with serological evidence of syphilis but without any clinical neurological symptom/sign. This form could appear weeks after infection;
-   meningeal form: characterized by diffuse meninges inflammation that could lead to headache, nausea, vomiting, neck stiffness, photophobia, cranial nerve deficits (including deafness, vertigo and blindness), confusion, lethargy and seizures. This form could appear weeks/months after infection.

Early or late neurosyphilis (1–10 years after the primary infection):

-   meningovascular form: inflammation of the meninges and a vasculitis of small and medium arteries happen, leading to thrombosis, causing strokes as well as myelopathy if the spinal cord vessels are involved.

Late neurosyphilis develops decades after the primary infection and is also known as the parenchymal form. It is caused by chronic meningeal reaction to spirochetal invasion. It develops in 10–20% of cases and two different forms are further recognized:

-   general paralysis of the insane/paralytic dementia: caused by a chronic meningoencephalitis leading to diffuse cerebral atrophy. Clinically it could be characterized by the appearance of mood disturbances, personality changes, psychosis, mania, dementia with memory and executive function impairment, confusion, seizures, tremors, dysarthria and pupillary abnormalities (known as Argyll Robertson pupils);
-   tabe dorsalis: caused by the degeneration of the dorsal columns and roots of the spinal cord, leading to sensory ataxia and crisis of acute pain, but also bladder dysfunction and pupillary and ocular palsies.

Seizures can happen at any stage during neurosyphilis. The incidence of acute symptomatic seizures due to neurosyphilis is extremely variable and could range from 14 to 60% [5]. Status epilepticus (SE) is considered a rare manifestation of neurosyphilis and until now few reports or small case series have been published. The extreme heterogenicity, as well as the extreme time lapse between the first syphilitic infection and the first seizure, make its diagnosis extremely challenging. Nevertheless, its prompt recognition and diagnosis enable the appropriate antibiotic therapy to be started rapidly alongside the anti-seizure treatment, thus increasing the chances of a good outcome.

SE could also represent the manifestation of Jarisch–Herxheimer reaction (JHR), a transitory worsening of clinical conditions that is probably immune-mediated and self-limited, mostly manifesting in the first 24 h after the beginning of the antibiotic therapy [1,6,7].

The possible pathogenesis involves lipoproteins, that probably trigger inflammatory reactions. As a matter of fact, in patients with JHR, elevated serum levels of cytokines have been found. As Rissardo et al. supposed [8], these inflammatory findings could also have a role in the pathogenesis of SE during JHR.

We present a case of a neurosyphilis-related new onset refractory status epilepticus (NORSE) [9] and a review of the existing literature on status epilepticus in neurosyphilis infection. Since familiarity with neurosyphilis has significantly decreased among clinicians, the aim of the present work is to give a comprehensive review of SE in this condition to improve the diagnostic workflow and to start rapid treatment.

## 2. Case Report

A 55-year-old man was admitted to our hospital for the acute onset of repeated seizures characterized by staring and oral automatisms followed by aphasia. The previous medical history was unremarkable. The patient's sister reported that, in the previous two months, he complained about gastric discomfort, anorexia with weight loss and blurred vision. At hospital admission the patient was afebrile. Neurological examination showed only the absence of myotatic reflexes of the lower limbs. Brain CT and CT-angiography scans were normal. Seizures rapidly worsened, becoming continuous. An urgent EEG showed continuous paroxysmal activity arising from the right temporal leads with bilateral diffusion. Thus, a diagnosis of focal non-convulsive status epilepticus (NCSE) with impaired awareness was made (Figure 1).

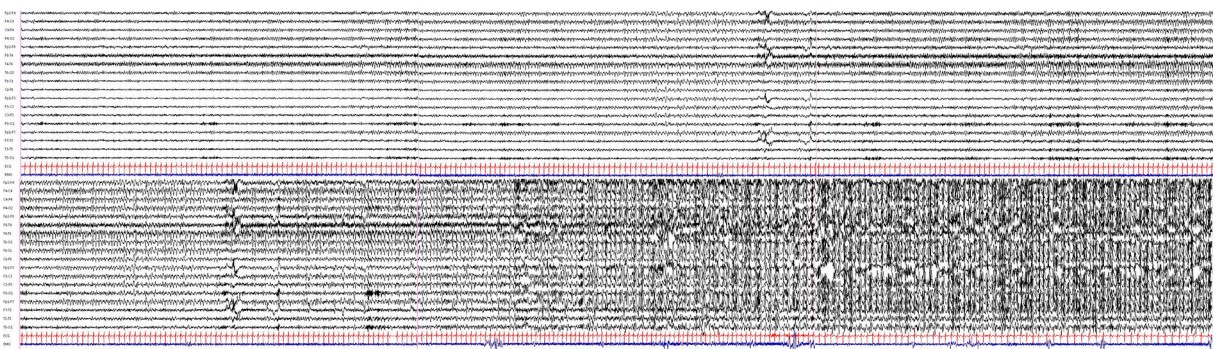

**Figure 1.** Ictal EEG (60 s/page). Rhythmic theta activity (5 Hz) evolving into delta activity (2–3 Hz) with sharp components arising from the right temporal lobe (T4-T6) then spreading to the ipsilateral parietal and frontal lobes. After 180 s a bilateral involvement of the frontal lobes appears. Then, a morphology evolution toward a spike and wave and poli-spike and wave discharges at 2 Hz is visible, and the epileptic activity becomes diffuse with a frontal predominance. The seizure ends abruptly after 8 min.

The patient received i.v. diazepam (10 mg) followed by i.v. valproic acid (bolus of 1200 mg; maintenance 2400 mg/day) and lacosamide (200 mg bolus; maintenance 400 mg/day), however, the SE did not resolve, and he was therefore admitted to the intensive care unit (ICU) and anesthetic therapy under continuous EEG monitoring (CEEG) was started. Blood analysis and urinary toxicology were unrevealing. CSF examination showed two cells/μL and a mild protein increment (93 mg/dl); viral PCR (HSV1-2, CMV, EBV, VZV, parvovirus B19, enterovirus, morbillivirus and mumps virus) and tests for bacteria were negative as well as glucose levels, while the immune-electro-focusing showed thirty intrathecal oligoclonal bands. At this stage, a diagnosis of cryptogenic NORSE was given.

In the ICU the patient was initially treated with midazolam plus propofol to obtain a stable burst suppression pattern. At the weaning of anesthetics after 24 h, SE recurred and ketamine (bolus dose of 100 mg, followed by a maintenance dose of 5 mg/Kg/h for 72 h) was started. Four days later, at the second weaning of anesthetic therapy, SE did not recur, and the patient continued only on i.v. anti-seizure medications.

Brain MRI (acquired within 24 h from SE onset) showed a mild ventricular enlargement and bilateral hippocampal atrophy (Figure 2).

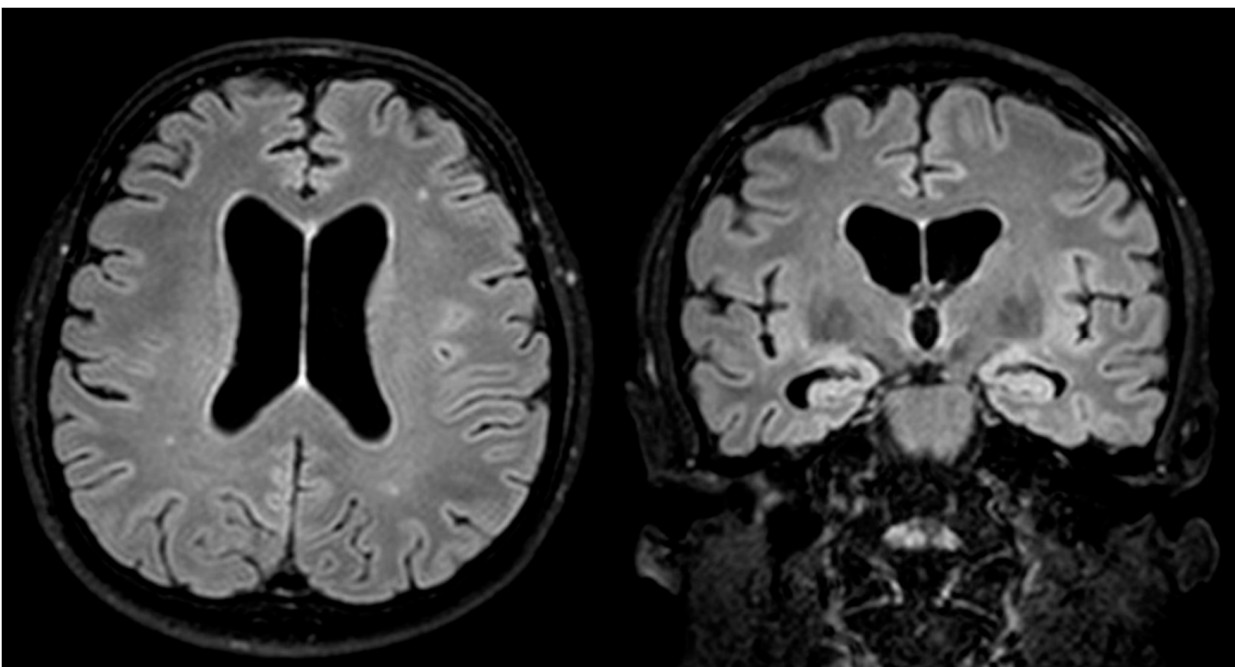

**Figure 2.** Brain MRI. Axial and coronal FLAIR brain MRI acquired on the first day of SE. MRI showed mild bilateral ventricular enlargement and hippocampal volume loss (right > left) without overt hyperintensities, and neither contrast enhancement nor leptomeningeal involvement.

Serum and CSF autoimmune encephalitis and onconeural antibodies (anti-GAD, anti-Yo, Ri, Hu, anti-Ma2, N-methyl-D-aspartate receptor (NMDAR), leucine-rich glioma inactivated protein 1 (LGI1), contactin-associated protein-like 2 (CASPR2)), as well as a serum autoimmune panel (antinuclear antibodies (ANAs), anti-phospholipid antibodies, anti-DNA antibodies, anti-cardiolipin antibodies, anti-extractable nuclear antigen antibodies (anti-ENAs) and anti-thyroid antibodies) and neoplastic markers (CEA, CA-19.9, AFP, PSA, CA125, NSE) were negative.

Eight days after SE onset, syphilis serology was positive in blood and CSF: the serum venereal disease research laboratory test (VDRL) had a title of 1 : 32 and a treponema pallidum hemagglutination assay (TPHA) had 1 : 5120. CSF IgM and IgG for Treponema pallidum were detected and RPR was 1 : 4. Serology for HBV, HCV and HIV, as well as serology for other sexually transmitted diseases (STDs), was negative. Thus, a diagnosis of neurosyphilis was made and therapy with i.m. penicillin G (procaine penicillin G 2.4 million units i.m. once daily and probenecid orally four times a day for 14 days) was started. In the immediate 12 h after the first penicillin G administration the patient experienced a transitory worsening characterized by the development of fever, hypertension and psycho-motor delirium with complex visual hallucinations which was interpreted as a JHR. During the following days, after maintenance of the antibiotic therapy, the patient slowly but fully recovered.

At the last clinical follow-up, 33 months after the SE development, the patient was still seizure-free, administration of valproic acid was interrupted and lacosamide reduced to 100 mg/day. The patient did not develop cognitive sequelae and was able to return to work in his former occupation.

The patient gave written informed consent for publication.

### 3. Literature Review

*3.1. Material and Methods: Literature Search Strategy and Study Selection Process*

Full-text articles and conference proceedings were selected from a comprehensive search of PubMed, Web of Science, Medline and Scopus databases. We performed a literature search of cases of status epilepticus in the context either of a previous diagnosis of syphilis or neurosyphilis or as the first manifestation of neurosyphilis.

Keywords and their synonyms were combined in each database as follows: ("status epilepticus") AND ("syphilis" OR "neurosyphilis" OR "lue" OR "Jarisch–Herxheimer Reaction"). No filter was applied on the publication date of the articles, and all results of each database were included up to March 2021. After removal of duplicates, all articles were independently evaluated through a screening of title and abstract by two independent reviewers (G.G., S.M.). The same reviewers performed an accurate reading of all full-text articles, which were assessed for eligibility for this study, and executed data collection to minimize the risk of bias. Moreover, the references of the included articles were screened to search for other previously unincluded articles.

In case of disagreement among investigators regarding inclusion and exclusion criteria, the senior investigator (S.M.) made the final decision.

Inclusion criteria were:

1.  Definite diagnosis of status epilepticus.
2.  Definite diagnosis of syphilis.
3.  Articles written in the English language or papers written in other languages for which a detailed and clear English abstract was available.
4.  Papers published in a peer-reviewed journal.

Exclusion criteria were:

1.  Studies conducted on animals or in vitro models.
2.  Reviews, books.
3.  A previous diagnosis of symptomatic epilepsy or idiopathic/genetic epilepsy.

Data Extraction Process

The aforementioned research strategy identified 253 records (Figure 3).

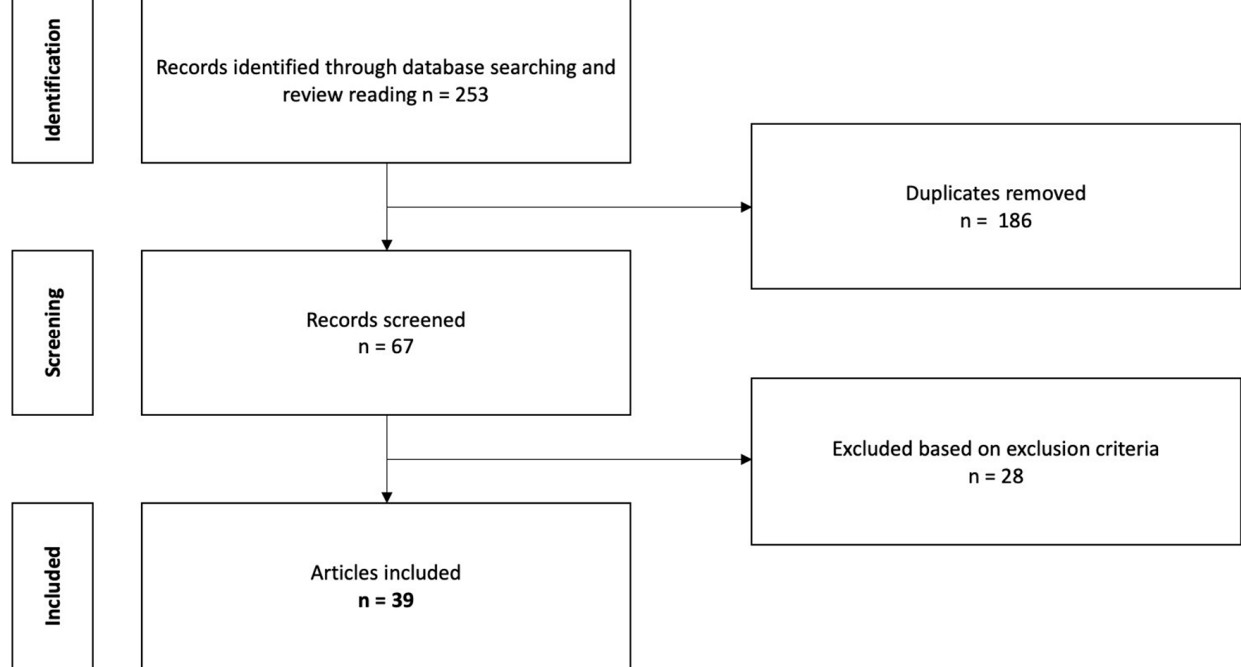

**Figure 3.** Flow-chart for literature review.

At the end of the selection process, 39 articles that satisfied inclusion criteria [8,10–46] were included in the study. Among them, we also included 4 articles [17,18,27,44] retrieved after examination of the references of all included articles.

## 4. Results

Overall, adding our patient to the already described ones, 50 patients with SE as neurosyphilis expression have been reported in the literature (see Supplementary Materials for details). In the work of Toudou-Daouda et al. (2018) [43], five patients were described but just for one of them was it clearly stated that SE was related to neurosyphilis, thus just one case was considered for the present review.

Table 1 shows the characteristics of patients who developed SE as a manifestation of neurosyphilis and of those with SE as an expression of JHR.

**Table 1.** Characteristics of neurosyphilitic patients presenting status epilepticus as an expression of neurosyphilis and of those presenting status epilepticus after antibiotic treatment initiation as an expression of a Jarisch–Herxheimer reaction.

| Clinical Characteristics | | SE as Expression of Neurosyphilis | SE as Expression of JHR in Neurosyphilis | *p* |
|---|---|---|---|---|
| **N** | | **N 45 (100%)** | **N 5 (100%)** | |
| Gender | Male | 33 (73%) | 4 (80%) | |
| | Female | 4 (9%) | 1 (20%) | 0.49 |
| | Not Reported | 8 (18%) | 0 | |
| Age at SE presentation (years) | Average/Median/Range | 46/45/29–62 | 48/48/23–71 | 0.762 |
| Immunodeficiency | Yes | 2 (5%) | 0 | |
| | No | 24 (53%) | 4 (80%) | 1.00 |
| | Not reported | 19 (42%) | 1 (20%) | |
| Clinical presentation | NOSE/NORSE | 35 (78%) | 4 (80%) | |
| | Epilepsy diagnosis before SE | 7 (16%) | 1 (20%) | 1.00 |
| | Not reported | 3 (7%) | 0 | |
| | Mood disturbances before SE | 8 (18%) | 2 (40%) | 0.26 |
| | Cognitive disturbances before SE | 14 (31%) | 4 (80%) | 0.05 |
| Syphilis diagnosis | At SE presentation | 38 (85%) | 4 (80%) | |
| | Already known syphilitic patients | 7 (16%) | 1 (20%) | 1.00 |
| Clinical types of SE | Convulsive forms | 21 (47%) | 1 (20%) | |
| | Non-convulsive forms | 19 (42%) | 4 (80%) | 0.35 |
| | Not reported | 5 (11%) | 0 | |
| SE response to treatment | Responsive | 21 (47%) | 3 (60%) | |
| | Refractory/Super-Refractory | 10 (22%) | 1 (20%) | 1.00 |
| | Not reported | 14 (31%) | 1 (20%) | |
| Neuroimage characteristics | Normal | 0 | 1 (20%) | 0.1 |
| | Atrophy | 9 (20%) | 3 (60%) | 0.08 |
| | White matter hyperintensities | 20 (44%) | 2 (40%) | 1.00 |
| | Strokes | 2 (5%) | 0 | 1.00 |
| | Gumma | 1 (2%) | 0 | 1.00 |
| | Not reported | 13 (29%) | 0 | 0.31 |
| Outcome | Recovery | 8 (18%) | 1 (20%) | 1.00 |
| | Improvement with sequelae | 16 (36%) | 1 (20%) | 0.65 |
| | Severe disability | 7 (16%) | 3 (60%) | **0.048** |
| | Deceased | 2 (5%) | 0 | 1.00 |
| | Not reported | 12 (25%) | 0 | 0.32 |

Note: in bold values statistically significant (*p* < 0.05)

Among the 50 patients, the great majority were males (37 (88%), five were females, in eight gender was not reported). Only two patients were clearly stated to be immunocompromised. The median age at SE presentation was 45 years (SD ± 10 years; age range 23–71).

In most of the cases (42 out of 47 (89%)), the SE determined the diagnosis of syphilis. In fact, only four patients already had a diagnosis of primary syphilis [31,39,41,44] and one had a diagnosis of fronto-temporal dementia (FTD) as an expression of late neurosyphilis [21].

Thirty-nine patients (83%) presented as new onset status epilepticus and, among them, ten patients had a refractory SE (three of them had a super-refractory status epilepticus). In eight cases, SE developed in the context of an already diagnosed epilepsy without a known etiology (cryptogenic), in whom the development of the SE episode was the event that finally led to the neurosyphilis diagnosis.

In five patients, SE developed 10 to 14 h after penicillin treatment initiation, thus was interpreted as part of a JHR (Table 1). It is not possible to find real differences in SE characteristics between this group and that of patients with SE not related to a JHR.

A history of subacute personality changes, mood disturbances, progressive memory impairment, general malaise (headache, weight loss, vertigo, anorexia) or undefined and subtle neurologic deficits (ataxia, blurred vision, speech disturbances) preceded SE development by days or weeks in most of the patients.

A motor SE, either generalized convulsive or focal motor, was reported for 22 patients, while NCSE (described as complex partial status epilepticus, CPSE) was reported for 23 patients (for five patients, the semeiology of SE was not clearly reported). The EEG showed a focal SE in 26 patients, mostly involving frontal and temporal lobes. Lateralized periodic discharges were clearly reported in 19 patients.

The MRI acquired in the acute phase of SE frequently showed T2/FLAIR hyperintensities involving the medial temporal structures and the frontal lobes (22 patients), frequently bilaterally, and sometimes with contrast enhancement.

Considering together the clinical presentation of subacute mood and memory impairment, repeated seizures and MRI alterations involving bilaterally the medial temporal lobes, the most frequent initial diagnosis was that of a limbic encephalitis. Whenever reported, these acute alterations showed disappearance after treatment in the follow-up MRIs. Strokes related to a syphilis's meningovascular form were reported in two patients [15,18], while a syphilitic gumma [13] was present in one case. Cerebral atrophy as the sole alteration was reported in eight patients.

CSF analysis showed mild increase in cellular count and proteins in most of the patients. The presence of intrathecal IgG oligoclonal bands was reported in three cases [20,23,42].

Follow-up information was available for 38 patients: nine patients had a complete recovery, 17 patients an improvement without a full recovery and ten patients presented severe cognitive and motor disability, while two patients died. Among the 39 patients with new onset SE, only two subsequently developed epilepsy. On the other hand, among the eight patients with an already known cryptogenic epilepsy, three patients experienced recurrent SE episodes until the diagnosis of neurosyphilis, leading to proper treatment.

## 5. Discussion

We reported a case of a middle-aged man developing a new onset refractory status epilepticus, finally diagnosed as being caused by neurosyphilis. NORSE has been defined recently as a clinical presentation, not a specific diagnosis, in a patient without active epilepsy or other pre-existing relevant neurological disorders, with new onset of refractory status epilepticus without a clear acute or active structural, toxic or metabolic cause [38]. NORSE generally carries a bad prognosis, but defining its etiology could have important prognostic and therapeutic implications, especially regarding those etiologies for which a specific treatment exists. Nevertheless, for half of the NORSE cases, etiology remains unknown even after a well-conducted and complete diagnostic work-up. Whenever a cause is identified, in up to 40% it is an inflammatory/autoimmune or paraneoplastic etiology and in up to 10% it is an unusual infective disease, while genetic, metabolic and toxic causes are considered rare [47]. Syphilis has been reported among NORSE infective etiologies and its prompt recognition and treatment can positively influence the outcome.

At the end of the 20th century, after the advent of penicillin, rates of syphilis reached a nadir. Since then, they have steadily climbed due to an increased prevalence of immuno-suppression, due to HIV infection and chronic immunomodulating treatments. Currently, even if it is difficult to exactly define its real incidence, in a previous paper addressing the clinical spectrum of neurosyphilis [4], the yearly incidence of neurosyphilis was estimated at about 0.2–2.1 cases per 100,000 inhabitants. The clinical presentation of neurosyphilis can be extremely heterogeneous and subtle, appearing at any stage, often many years or even decades after the primary infection. Indeed, clinical history of revised cases frequently revealed the presence of unrecognized subtle signs such as subacute personality changes, mood disturbances, progressive memory impairment, general malaise (with headache, anorexia and weight loss) and subtle neurologic deficits (e.g., ataxia/vertigo, blurred vision, speech disturbances). For all these reasons, it goes frequently unrecognized. Moreover, the primary infection itself is frequently self-limited and may also have passed unrecognized. Epilepsy can appear at any stage of neurosyphilis and its incidence has a great variability, while SE is considered a rare event.

The reviewed literature allowed us to recognize that the most frequent clinical scenario is represented by a fronto-temporal non-convulsive new onset status epilepticus that develops in a middle-aged, immunocompetent man without a known history of primary or secondary syphilis. Indeed, a de novo SE was the first clinically recognized manifestation of neurosyphilis in about 80% of the cases. However, it is important to note that in some cases SE developed in a patient with a previous diagnosis of epilepsy of undetermined origin and the SE occurrence led to an etiological diagnosis of neurosyphilis. These cases are paradigmatic: the clinician often does not consider this diagnostic possibility in cases of cryptogenic epilepsy. On the contrary, a crucial point concerns the importance to exclude this etiology even in immunocompetent patients, especially in a context with electro-clinical signs of temporal lobe involvement and in particular if associated with behavioral and memory alterations.

Indeed, the differential diagnosis between viral and neurosyphilitic limbic encephalitis can be challenging in some instances. In particular, neurosyphilis enters in differential diagnosis with HSV encephalitis. Saunderson and Chan [48] performed a literature search of cases of neurosyphilis presenting MRI meso-temporal lobe alterations undistinguishable from those found in HSV encephalitis. They reported 24 patients that had neuroradiological, clinical presentation and CSF findings suggestive for HSV encephalitis, but in the end, CSF PCRs were negative for HSV while treponemal and non-treponemal tests for syphilis were positive.

From the neuroradiological point of view, although the review of the literature does not provide specific elements, apart from the cases characterized by gummy lesions, it does allow us to outline a profile of MRI alterations, with a preferential involvement of the limbic, fronto-temporal regions. In the acute phases of the SE, the brain MRI can reveal extensive T2/FLAIR hyperintensities widely and mostly involving the medial temporal structures and the frontal lobes, that usually disappear in the follow-up MRIs acquired after the resolution of the SE and the application of appropriate anti-epileptic and antibiotic treatment. It is difficult to establish whether these acute alterations are related to the syphilis infection itself or represent, at least partially, a peri-critical acute modification [49] related to the sustained seizure activity. Furthermore, a diffuse atrophy is a common finding and could be related either to a chronic meningitis alone or to the coexistence of the chronic meningitis and seizure recurrence [50] and the chronic use of anti-seizure medication in patients with a previous diagnosis of epilepsy [51,52].

Regarding the acute phase of SE, treatment with antibiotics and ASMs resulted in resolution of the seizures in all cases. Approximately 30% of the SE cases were refractory (or super-refractory), which is in line with the response generated by treatment with ASMs in the general population of SE [53]. This finding confirms the importance of early etiological diagnosis for treatment. Moreover, the prompt application of the appropriate antibiotic and anti-epileptic therapy can determine a complete, or at least a partial, clinical recovery

in nearly 70% of cases, as in our patient. Lastly, it is interesting to note that among patients with a de novo status epilepticus, the development of a subsequent epilepsy was reported in only two cases. These data, although to be interpreted with caution, would indicate that seizures and SE in the context of neurosyphilis are acute symptomatic events and therefore have a low risk of recurrence once the treponema infection has been treated.

## 6. Conclusions

In the presence of either epilepsy of unknown etiology or a new onset status epilepticus, especially if they appear in a middle-aged man together with a history of subtle and progressive mood/cognitive impairment suggesting a limbic encephalitis-like presentation [54], it is recommended to investigate neurosyphilis rapidly. The present review highlights that it must be ruled out even in non-immunocompromised patients. In particular, it is crucial to have a high degree of clinical suspicion and its investigation should always be part of the extensive diagnostic work-up of NORSE. Prompt recognition of neurosyphilis and initiation of appropriate antibiotic therapy could partially or completely reverse neurologic sequelae, thus changing the natural progression of the disease.

**Supplementary Materials:** The following are available online at https://www.mdpi.com/article/10.3 390/neurosci2040031/s1, Table S1. Demographic, clinical, EEG, MRI, CSF and follow-up characteristics of patients presenting status epilepticus as an expression of neurosyphilis. Table S2. Demographic, clinical, EEG, MRI, CSF and follow-up characteristics of neurosyphilitic patients presenting status epilepticus after antibiotic treatment initiation as an expression Jarisch–Herxheimer reaction.

**Author Contributions:** G.G.: study concept and design, acquisition of data, analysis and interpretation of data, drafting the manuscript. S.M.: study concept and design, analysis and interpretation of data, study supervision, final approval of the version to be submitted. All authors have read and agreed to the published version of the manuscript.

**Funding:** This research received no external funding.

**Institutional Review Board Statement:** Not applicable.

**Informed Consent Statement:** Full and detailed consent from the patient was taken. The patient's identity has been adequately anonymized.

**Data Availability Statement:** Not applicable.

**Conflicts of Interest:** The authors have no conflict of interest to declare that are relevant to the content of this article.

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
