# Peer review of "Status Epilepticus and Neurosyphilis: A Case Report and a Narrative Review"

_neurosci, doi:10.3390/neurosci2040031_

Round 1

Reviewer 1 Report

I like the overall article. A good and smooth read. The authors reviewed and summarised previously published such cases and case series. I do not have any major comments, just a few minor suggestions as follows. 

  1. Line 41 - Did you place this patient-on-patient continuous EEG? If not, how can you be sure that SE was not resolved since they were non-convulsive? Please mention continuous EEG.
  2. Line 5-9: please elaborate on how long did you keep the patient in burst suppression for the first and second time?
  3. Line 43 – What was the differential on CSF WBCs? CSF glucose level? Did you obtain CSF culture?
  4. During the pharmacological coma, did you readjust AEDs? If so, worth mentioning here.
  5. Your patient had just 2 WBC in CSF with mild protein elevation which is relatively benign-looking CSF for infectious etiology. Did the authors find any such presentations? If so, should be worth mentioning in results and/or discussion.
  6. In conclusion section – line 24 – instead of ‘it should be mandatory’, I recommend using ‘It is advisable’ or ‘recommended’ or ‘suggest based on this article'.

Author Response

I like the overall article. A good and smooth read. The authors reviewed and summarised previously published such cases and case series. I do not have any major comments, just a few minor suggestions as follows. 

R: We thank the reviewer for the overall positive judgement.

  1. Line 41 - Did you place this patient-on-patient continuous EEG? If not, how can you be sure that SE was not resolved since they were non-convulsive? Please mention continuous EEG.

R: Thanks for the comment. First, an urgent EEG that let us diagnose SE was acquired in the emergency department. Then, the patient was transferred to the ICU and here anaesthetic therapy targeted to burst suppression under a continuous EEG monitoring (CEEG) was started. We changed the text (page 3, line 41).

  1. Line 5-9: please elaborate on how long did you keep the patient in burst suppression for the first and second time?

R: We added the BS time for the first anesthetic cycle in the text (page 4, line 5).

  1. Line 43 – What was the differential on CSF WBCs? CSF glucose level? Did you obtain CSF culture?

R: CSF WBC were 2/mm3 , CSF glucose levels were normal and CSF culture was negative (page 4, line 2).

  1. During the pharmacological coma, did you readjust AEDs? If so, worth mentioning here.

R: Thanks for the comment. No, during therapeutic coma the AEDs therapy was maintained stable.

  1. Your patient had just 2 WBC in CSF with mild protein elevation which is relatively benign-looking CSF for infectious etiology. Did the authors find any such presentations? If so, should be worth mentioning in results and/or discussion.

R: We thank the reviewer for the comment. Neurosyphilis is usually accompanied by CSF pleocytosis which declines over decades. CSF pleocytosis is generally mild compared to other CNS infectious disease and it could vary a lot (Gonzalez et al Semin Neurol 2019). Among the reported cases, three had very mild CSF WBC levels (< 10/mm3). To note, all these three patients as in our case have a good outcome with improvement/recovery. Differently from our case, they presented a SE episode responsive AEDs therapy without the need to anesthetic therapy to be administered.

  1. In conclusion section – line 24 – instead of ‘it should be mandatory’, I recommend using ‘It is advisable’ or ‘recommended’ or ‘suggest based on this article'.

R: Thanks for the comment. We changed the text (page 9, line 24).

Reviewer 2 Report

This is a well written and comprehensive case report for a very important topic.

Author Response

This is a well written and comprehensive case report for a very important topic.

R: We thank the reviewer for the positive judgement on our study.

Reviewer 3 Report

The manuscript focuses on status epilepticus in neurosyphilis as a rare, however, clinically relevant manifestation. The manuscript is well written, and could profit from a moderate revision of English language. 

I have a few points and remarks. 

  1. It would be nice to have a short summary on further clinical symptoms and presentation forms of neurosyphilis (like tabes, strokes, progressive paralysis, aphasia, etc.) and of their relative occurrence frequencies in the Introduction, as SE is apparently a rare form of manifestation.
  2. Table 1 deals with SE related to Neurosyphilis per se, and Table 2 with SE related to Jarisch-Herxheimer reaction. However, it is a bit confusing that in the text You find different numbers not matching those in Table 1. It took me some time to - hopefully - understand that Table 1 does not deal wit the few cases related to JHR at all. For the clear presentation it would be important to draw attention to this fact (I hope to have understood it correctly).
  3. Apparently - at least according to the numbers and the supporting material - in some cases it was not documented whether SE was responsive or refractory, or what exactly the outcome was, etc. I suggest to indicate the number of missing cases (N/R in supplementary. material) also in Table 1/2 for the respective variables, again, for the clear presentation.

Author Response

The manuscript focuses on status epilepticus in neurosyphilis as a rare, however, clinically relevant manifestation. The manuscript is well written, and could profit from a moderate revision of English language. I have a few points and remarks. 

R: We thank the reviewer for the overall positive comments.

  1. It would be nice to have a short summary on further clinical symptoms and presentation forms of neurosyphilis (like tabes, strokes, progressive paralysis, aphasia, etc.) and of their relative occurrence frequencies in the Introduction, as SE is apparently a rare form of manifestation.

R: We thank the reviewer for this comment. We added in the introduction a short paragraph describing the other clinical manifestations of Neurosyphilis (page 2, lines 24-54).

  1. Table 1 deals with SE related to Neurosyphilis per se, and Table 2 with SE related to Jarisch-Herxheimer reaction. However, it is a bit confusing that in the text You find different numbers not matching those in Table 1. It took me some time to - hopefully - understand that Table 1 does not deal wit the few cases related to JHR at all. For the clear presentation it would be important to draw attention to this fact (I hope to have understood it correctly).

R: We thank the reviewer for the comment. To clarify we replaced table 1 and 2 creating just one table with SE during neurosyphilis and those related to JHR in neurosyphilitic patients during antibiotic therapy. The numbers and percentages reported in the text are related to the entire cohort.

  1. Apparently - at least according to the numbers and the supporting material - in some cases it was not documented whether SE was responsive or refractory, or what exactly the outcome was, etc. I suggest to indicate the number of missing cases (N/R in supplementary. material) also in Table 1/2 for the respective variables, again, for the clear presentation.

R: We thank the reviewer for the comment. We changed the table inserting the numbers of not reported patients.

Reviewer 4 Report

The manuscript presented by Giada Giovannini  and Stefano Meletti is highly interesting and well written. The topic is original and the author presented in detail the case report. However I would suggest the authors to improve the introduction, to include also (if are presented in literature) previous studies that showed the pato-physiology difference between Neurosyphilis and Syphilis in general (because is not clear by the introduction).

Author Response

The manuscript presented by Giada Giovannini and Stefano Meletti is highly interesting and well written. The topic is original and the author presented in detail the case report. However I would suggest the authors to improve the introduction, to include also (if are presented in literature) previous studies that showed the pato-physiology difference between Neurosyphilis and Syphilis in general (because is not clear by the introduction).

R: We thank the reviewer for this comment. We added to the introduction a brief paragraph on reported pathogenetic mechanisms (page 1, lines 36-42; page 2, lines 1-20).